REGISTERED REPORT PROTOCOL

# The value of TI-RADS combined with superb microvascular imaging in distinguishing thyroid nodules: A protocol for systematic review and meta-analysis

Cong Wang[1], Mingxin Lin[1], Lin Zhong[2]*, Congliang Tian [3]*

1 Ultrasound Department of the First Affiliated Hospital of Dalian Medical University, Dalian, China,
2 Pathology Department of the First Affiliated Hospital of Dalian Medical University, Dalian, China,
3 Pediatrics Department of the First Affiliated Hospital of Dalian Medical University, Dalian, China

☯ These authors contributed equally to this work.
* 365213427@qq.com (LZ); fytgzy@163.com (CT)

## Abstract

### Background

As a novel ultrasonic technique, superb microvascular imaging (SMI) can quickly, simply and noninvasively observe the microvascular distribution in a tumor and evaluate the microvascular perfusion. Previous studies have shown that SMI can detect the blood flow signals of neovascularization in tumors and increase the sensitivity for detecting thyroid cancer. However, the results of these studies have been contradictory, and the sample sizes were too small. Therefore, the present meta-analysis will aim at evaluating the value of a thyroid imaging report and data system (TI-RADS) combined with SMI in distinguishing between benign and malignant thyroid nodules.

### Methods

We will search PubMed, Web of Science, Cochrane Library, Google Scholar, and Chinese biomedical databases from their inceptions to the June 31, 2020. Two authors will independently carry out searching literature records; scanning titles, abstracts, and full texts; collecting data; and assessing risk of bias. Review Manager 5.2 and Stata14.0 software will be used for data analysis.

### Results

This systematic review will evaluate the value of TI-RADS combined with SMI in distinguishing between benign and malignant thyroid nodules.

### Systematic review registration

INPLASY202070113.

**Data Availability Statement:** All relevant data from this study will be made available upon study completion.

**Funding:** This study is supported by Liaoning Natural Science Foundation Project (20170540256). The funders had no role in study design, data collection and analysis, decision to publish, or preparation of the manuscript.

**Competing interests:** The authors have declared that no competing interests exist.

## Introduction

Thyroid cancer is a very common malignant disease that accounts for approximately 1% of all cancer patients [1]. A solid thyroid nodule is a risk factor for thyroid cancer, so it is very important to differentiate thyroid nodules effectively and in a timely manner [2]. Ultrasonography has the advantage of high sensitivity in the diagnosis of a thyroid nodule, and it is the first choice for the clinical diagnosis and differentiation of thyroid cancer [3]. In 2017, the American College of Radiology (ACR) proposed the latest version of thyroid imaging report and data system (TI-RADS) classification, ACR TI-RADS, based on large-scale, evidence-based clinical validation, which greatly promotes the differentiation of benign and malignant thyroid nodules [4]. However, due to the complexity and overlapping of the sonograms of thyroid nodules, it is still difficult to accurately identify some nodules with atypical ultrasound characteristics [5]. There are some differences in blood flow pattern and vascular morphology between benign and malignant thyroid nodules, which is helpful in distinguishing between benign and malignant thyroid nodules [6]. Color Doppler flow imaging (CDFI) is often used to show the blood flow inside the tumor, but CDFI is not good for some low-velocity microvessels and is not useful in distinguishing thyroid nodules [7,8]. As a novel ultrasonic technique, SMI can quickly, simply and noninvasively observe the microvascular distribution in the tumor and evaluate the microvascular perfusion [9]. Previous studies have shown that SMI can detect the blood flow signals of neovascularization in tumors and increase the sensitivity for detecting thyroid cancer [10–13]. However, the results of these studies have been contradictory, and the sample sizes were too small. Therefore, the present meta-analysis aimed at evaluating the value of TI-RADS combined with SMI in distinguishing between benign and malignant thyroid nodules.

## Materials and methods

This study will be conducted in accordance with the Preferred Reporting Items for Systematic Reviews and Meta-Analyses (PRISMA) guidelines, and the protocol was registered in the INPLASY (INPLASY202070113).

### Eligibility criteria

**Type of study.** This study will only include high-quality clinical cohorts or case control studies.

**Type of patients.** The patients had all been diagnosed with a thyroid nodule.

**Intervention and comparison.** Thyroid nodules of all patients were assessed with TI-RADS alone and with TI-RADS combined with SMI.

**Type of outcomes.** The primary outcomes include the area under the curve (AUC) of the summary receiver operating characteristic (SROC) curve, as thyroid nodules were assessed by means of TI-RADS alone and TI-RADS combined with SMI.

### Search methods

PubMed, Web of Science, Cochrane Library, Google Scholar and Chinese biomedical databases will be searched from their inceptions to the June 31, 2020. The search strategy for PubMed is shown in Table 1. Other online databases will be used with the same strategy.

### Data extraction and quality assessment

Two authors will independently selecte the trials according to the inclusion criteria and imported them into Endnote X9. Then, the duplicated or ineligible studies will be removed.

**Table 1. Search strategy sample of PubMed.**

| Number | Search terms |
|--------|--------------|
| 1 | thyroid cancer or thyroid neoplasm or thyroid tumor or thyroid nodule |
| 2 | thyroid imaging report and data system or TI-RADS |
| 3 | superb microvascular imaging or superb micro-vascular imaging |
| 4 | and 1–3 |

Afterwards, the titles, abstracts, and full texts of all the literature will be screened to identify eligible studies. All essential data will be extracted using the data collection sheet previously created by 2 independent authors. Discrepancies in data collection between the 2 authors will be settled through discussion with the help of another author. The following data will be extracted from each included study: the first author's surname, publication year, language of publication, study design, sample size, number of lesions, source of the subjects, "gold standard," and diagnostic accuracy. The true positives (TPs), true negatives (TNs), false positives (FPs), and false negatives (FNs) in the fourfold (2 x 2) tables will also be collected. Methodological quality will be assessed by two researchers independently based on the Quality Assessment of Diagnostic Accuracy Studies (QUADAS) tool [14]. The QUADAS criteria includes 14 assessment items. Each of these items was scored as "yes" (2), "no" (0), or "unclear" (1). The QUADAS score ranged from 0 to 28, and a score $\geq$ 22 indicated good quality.

## Statistical analysis

STATA version 14.0 (Stata Corp, College Station, TX, USA), Meta-Disc version 1.4 (Universidad Complutense, Madrid, Spain), and MedCalc version 15.2.2 (MedCalc Software, Ostend, Belgium) software will be used for meta-analysis. We will calculate the pooled summary statistics for sensitivity (Sen) and specificity (Spe) with their 95% confidence intervals (CIs). The summary receiver operating characteristic (SROC) curve and corresponding area under the curve (AUC) were obtained. We will compare the two AUCs of TI-RADS alone and TI-RADS combined with SMI. Cochran's Q-statistic and $I^2$ test will be used to evaluate potential heterogeneity between studies. If the Q-test showes a $P < 0.05$ or $I^2$ test exhibited $> 50\%$, indicating significant heterogeneity, the random-effects model will be employed, or if heterogeneity is not significant, the fixed-effects model will be used. We will conduct Begg's funnel plots and Egger's linear regression tests to investigate publication bias. Sensitivity analysis will be performed to evaluate the influence of a single study on the overall estimate.

## Ethics and dissemination

We will not obtain ethics documents because this study will be conducted based on the data of published literature. We expect to publish this study in a peer-reviewed journal.

## Discussion

Thyroid nodules are a common clinical disease, and their accurate differentiation has important guiding significance for clinical decision-making. High resolution ultrasound plays an important role in the differential diagnosis of thyroid nodules [13]. Although ultrasound features of malignant thyroid nodules, such as low echo, unclear margin, microcalcifications and aspect ratio $> 1$, increase the risk of evaluation of malignant nodules, no single ultrasound feature can independently diagnose malignant thyroid nodules [15]. Present studies have revealed that SMI was able to identify low-velocity blood flow without being affected by motion artifacts, and as an adjunct to grayscale ultrasound, SMI showed a significantly improved

diagnostic performance in differentiating between benign and malignant thyroid nodules [16]. The present meta-analysis aims at evaluating the value of thyroid imaging report and data system (TI-RADS) combined with SMI in distinguishing between benign and malignant thyroid nodules and to provide evidence on the evidence-based medical support for clinical practice.

## Supporting information

**S1 Checklist. PRISMA-P (Preferred Reporting Items for Systematic review and Meta-Analysis Protocols) 2015 checklist: Recommended items to address in a systematic review protocol**∗.
(DOC)

**S1 File.**
(PDF)

## Author Contributions

**Conceptualization:** Cong Wang.

**Data curation:** Congliang Tian.

**Formal analysis:** Lin Zhong.

**Investigation:** Cong Wang, Mingxin Lin.

**Software:** Cong Wang, Mingxin Lin.

**Validation:** Lin Zhong.

**Visualization:** Lin Zhong.

**Writing – original draft:** Congliang Tian.

**Writing – review & editing:** Cong Wang.

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
