## [Decision Letter · Decision Letter 0]

14 Sep 2020

PONE-D-20-23187

The value of TI-RADS combined with superb microvascular imagine in distinguishing thyroid nodules: A protocol for systematic review and meta-analysis

PLOS ONE

Dear Dr. Tian,

Thank you for submitting your manuscript to PLOS ONE. After careful consideration, we feel that it has merit but does not fully meet PLOS ONE’s publication criteria as it currently stands. Therefore, we invite you to submit a revised version of the manuscript that addresses the points raised during the review process.

We look forward to receiving your revised manuscript.

Kind regards,

Oded Cohen

Academic Editor

PLOS ONE

Journal Requirements:

2.Thank you for stating the following financial disclosure:

 [Initials of the authors who received each award. The funders  will not have a role in study design, data collection and analysis, decision to publish, or preparation of the manuscript.].

Additional Editor Comments (if provided):

The authors should use Google Scholar as a search engine or to explain why particular search engines were chosen

The headline refers to both systematic review and meta-analysis while the abstract refers to systematic review alone. As this are two distinct entities, the authors should clarify which type of work is about to be done

The authors must adhere to the PRISMA protocol for systematic review and meta-analysis. This should be stated and carries out throughout the manuscript.

The authors should explain better current knowledge on regular Doppler and why it was not adopted for ACR TIRADS, and then address differences of regular Doppler and SMI

The type of studies is not well defined. Please adhere to the PRISMA protocols

How can the authors define 'without language restriction'? How do the plan to overcome this barrier in order to fully understand the articles?

Why does criteria 1 in table 1 needed? Please explain

Discussion does contain a 'will be' which will address the study's result.  Is it just the opening statement?

---

## [Decision Letter · Decision Letter 1]

22 Dec 2020

The value of TI-RADS combined with superb microvascular imagine in distinguishing thyroid nodules: A protocol for systematic review and meta-analysis

PONE-D-20-23187R1

Dear Dr. Congliang Tian,

We’re pleased to inform you that your manuscript has been judged scientifically suitable for publication and will be formally accepted for publication once it meets all outstanding technical requirements.

Kind regards,

Oded Cohen

Academic Editor

PLOS ONE

Additional Editor Comments (optional):

Reviewers' comments:

Reviewer's Responses to Questions

**Comments to the Author**

1. Does the manuscript provide a valid rationale for the proposed study, with clearly identified and justified research questions?

Reviewer #1: Yes

Reviewer #2: Yes

2. Is the protocol technically sound and planned in a manner that will lead to a meaningful outcome and allow testing the stated hypotheses?

Reviewer #1: Yes

Reviewer #2: Yes

3. Is the methodology feasible and described in sufficient detail to allow the work to be replicable?

Reviewer #1: Yes

Reviewer #2: Yes

4. Have the authors described where all data underlying the findings will be made available when the study is complete?

Reviewer #1: Yes

Reviewer #2: Yes

5. Is the manuscript presented in an intelligible fashion and written in standard English?

Reviewer #1: Yes

Reviewer #2: Yes

6. Review Comments to the Author

You may also provide optional suggestions and comments to authors that they might find helpful in planning their study.

Reviewer #1: It sounds like an interesting issue, The pattern of the blood flow in a tumor might be a tool for more precise US diagnosis. good luck

Reviewer #2: The study aims to evaluate the value of a thyroid imaging report and data system (TI-RADS) combined with SMI in distinguishing between benign and malignant thyroid nodules.

The study protocol is well written.

There authors should relate to risk of bias and dealing with missing data in the data extraction and quality assessment section.

7. PLOS authors have the option to publish the peer review history of their article (what does this mean?). If published, this will include your full peer review and any attached files.

Reviewer #1: **Yes: **Galit Avior

Reviewer #2: **Yes: **Sharon Tzelnick

---

## [Editor Report · Acceptance letter]

28 Dec 2020

PONE-D-20-23187R1 

The value of TI-RADS combined with superb microvascular imaging in distinguishing thyroid nodules: a protocol for systematic review and meta-analysis 

Dear Dr. Tian:

I'm pleased to inform you that your manuscript has been deemed suitable for publication in PLOS ONE. Congratulations! Your manuscript is now with our production department. 

Kind regards, 

on behalf of

Dr. Oded Cohen 

Academic Editor

PLOS ONE